# Obesity as a Condition Determined by Food Addiction: Should Brain Endocannabinoid System Alterations Be the Cause and Its Modulation the Solution?

**DOI:** 10.3390/ph14101002

**Published:** 2021-09-29

**Authors:** Marialuisa de Ceglia, Juan Decara, Silvana Gaetani, Fernando Rodríguez de Fonseca

**Affiliations:** 1UGC Salud Mental, Instituto de Investigación Biomédica de Málaga (IBIMA), Universidad de Málaga-Hospital Universitario Regional de Málaga, 29010 Málaga, Spain; juan.decara@ibima.eu; 2Department of Physiology and Pharmacology “V. Erspamer”, Sapienza University of Rome, 00185 Rome, Italy; silvana.gaetani@uniroma1.it

**Keywords:** obesity, food addiction, endocannabinoid system, brain, reward system, cannabinoid receptors

## Abstract

Obesity is a complex disorder, and the number of people affected is growing every day. In recent years, research has confirmed the hypothesis that food addiction is a determining factor in obesity. Food addiction is a behavioral disorder characterized by disruptions in the reward system in response to hedonic eating. The endocannabinoid system (ECS) plays an important role in the central and peripheral control of food intake and reward-related behaviors. Moreover, both obesity and food addiction have been linked to impairments in the ECS function in various brain regions integrating peripheral metabolic signals and modulating appetite. For these reasons, targeting the ECS could be a valid pharmacological therapy for these pathologies. However, targeting the cannabinoid receptors with inverse agonists failed when used in clinical contexts as a consequence of the induction of affective disorders. In this context, new classes of drugs acting either on CB1 and/or CB2 receptors or on synthetic and degradation enzymes of endogenous cannabinoids are being studied. However, further investigation is necessary to find safe and effective treatments that can exert anti-obesity effects, normalizing reward-related behaviors without causing important adverse mood effects.

## 1. Introduction

At the time of writing, the number of obese people in the world is estimated to be around 2.1 billion, which represents 30% of the total adult population [1]. Unfortunately, this number rises every day. In the 1990s, the World Health Organization sounded the alarm, calling obesity a new pandemic disease [2]. At the same time, public awareness campaigns were initiated to sensitize policy-makers, private sector partners, medical professionals, and the public at large, but they did not result in effective actions or in slowing the diffusion of the pathology. Over time, the obesity epidemic spread, not only in industrialized societies but also in developing countries, thus transforming into a worldwide concern [2].

Obesity is a chronic pathology, and managing it is a long-term or life-long process [3]. Generally, obesity is characterized by excessive accumulation of body fat in various districts of the body. Clinically, the most used parameter to define overweight and obesity is the body mass index, or BMI, which is obtained by dividing a person’s weight in kilograms by their squared height in meters. BMI is the most used index to describe body mass and obesity, but it does not take many other factors into account, such as gender, age, or percentage of lean/fat mass [4]. For this reason, it is always accompanied by other measures, for example, the abdominal circumference, since the accumulation of fat in the visceral area is correlated to cardiovascular and metabolic disorders [5].

Multiple factors can influence the development of obesity. There is a complex relationship among biological, psychosocial, and behavioral components, including genetic makeup, socioeconomic status, and cultural influence [6]. In addition to these elements, obesity development has been linked to microorganisms, epigenetics, increased maternal age, lack of sleep, endocrine disruptors, pharmaceutical iatrogenesis, comorbid conditions, and their treatments [7].

Independent of the clinical reasons behind obesity development, hyperphagia, or chronic overeating, is a primary behavioral component of this pathology [8,9], and many forms of obesity can be considered a consequence of overeating [10]. This deregulated feeding behavior is enabled by the availability of calorie-dense foods [10,11], reduced physical activity due to a sedentary lifestyle, unhealthy eating habits, and increased exposure to stress [12,13].

Obesity is an important pathology to study due to its widespread diffusion and its harmfulness; it has been linked to a progressive increase in the risk of death in both men and women [14]. In fact, obesity is considered a risk factor for the development of numerous pathologies such as:Cardiovascular diseases, such as hypertension [15], myocardial infarction [16], and stroke [17];Metabolic disorders, such as type 2 diabetes [18] and chronic hyperglycemia [19];Dementia [20] and Alzheimer’s’ disease [21];Cancer, especially breast, endometrial [14], colon, and kidney [22];Hepatic dysfunction and cirrhosis [14];Pulmonary diseases such as sleep apnea and asthma [23];Osteoarthritis [24];Fertility, pregnancy, and delivery complications [25];Increase of anxiety, depression, and suicide [26].

In addition to the cited pathologies, obesity is associated with chronic low-grade inflammation [27] that participates in the link between obesity and comorbid diseases [28]. Over time, lipid accumulation in muscle, liver, and blood vessels activates immune responses, contributes to organ-specific disease, and exacerbates systemic insulin resistance [29]. Obesity causes tissue-specific proinflammatory changes; inflammation in pancreatic islets can reduce insulin secretion and trigger β cell apoptosis [30], and lipid accumulation in the liver causes nonalcoholic fatty liver disease, associated with an increase in cytokines and immune cells [31]. Additionally, obesity has been associated with increased inflammatory cytokine production and increased inflammation in skeletal muscle [32] and with chronic neuroinflammation, characterized by increased permeability of the blood-brain barrier, incremented production of inflammatory mediators, and glial cell activation [33].

For these reasons, it is possible to state that obesity is a heterogeneous disorder [34]. In contrast with what has been stated before, it is important to underline that recent studies have shown evidence of metabolically healthy obese patients (that is, obese people that present fewer metabolic abnormalities; for further information, see [35]). Healthy obesity is characterized by a reduction in quality of life; even though healthy obese people do not present any of the cited comorbidities, they may experience social stigma, difficulties in everyday tasks, and long-term impacts on the musculoskeletal system that could eventually hamper their quality of life. Healthy obesity clearly differs from compulsive obesity in that individuals keep overeating despite negative health consequences that lower their life expectancy.

As new mechanisms behind obesity development were discovered in recent years, researchers differentiated obesity into two different types: metabolic and hedonic [34,36]. Metabolic obesity is associated with impairments of the homeostatic control of food intake, whereas hedonic obesity can be considered more of a consequence of food addiction. The latter is characterized by disruptions of the reward system [36] that cause persistent overeating, an increased ratio between energy expenditure and body mass, comorbidity with eating disorders, depression, anxiety, stress, and sleep deprivation [34].

Currently, obesity still lacks an effective one-size-fits-all treatment, and hedonic obesity linked to food addiction is not considered a proper pathology, increasing the discrimination towards obese individuals. This is clinical nonsense if we consider the obesity-associated comorbidities that shorten life expectancy and require important medical intervention. This situation has been supported by the use of pharmacotherapies that only aim to treat two aspects of obesity: the central hypothalamic mechanisms that control appetite and the altered metabolic consequences derived from the disrupted metabolism (i.e., dyslipidemia or diabetes type 2). The discovery of the role of the endogenous cannabinoid system (ECS) as a potential modulatory system that might integrate all these homeostatic processes prompted a serious attempt to develop an integrative therapy for obesity (the Rimonabant in Obesity (RIO), discussed further on this work), addressing both hedonic and metabolic factors. Although that attempt failed because of unexpected psychiatric events, the lessons from the RIO studies showed the possibility of using the ECS as a source of integrative therapies for obesity. The aim of this review is to summarize the recent evidence published about food addiction and its physiopathology and to report the latest studies about the role of the ECS in both obesity and food addiction, with a particular focus on the brain-periphery interaction. Moreover, in the last part of the work, the attention will be centered on the newest drugs acting on ECS, developed with antiobesogenic activity after the RIO fiasco. Research in this field is still relevant, considering the role of ECS in controlling food intake and food reward-related processes.

## 2. Food Addiction

As stated above, one of the largest contributors to obesity is overeating, but why do people eat much more than they need for their energy expenditure? No clear answer for this seemingly simple question has been given. Since 1950, more and more studies have compared obesity and drug addiction since they are both characterized by repetitive consummatory behaviors that an individual is unable to control despite their awareness of undesirable consequences [37]. In 1956, Randolph first introduced the concept of food addiction, referring to a common pattern of symptoms observed in the consumption of foods such as corn, wheat, coffee, milk, eggs, and potatoes [38] similar to those of other addictive diseases. Moreover, researchers tried to establish a way to measure food addiction, developing the Yale Food Addiction Scale (YFAS) [39], a list of selected questions used to assess food addiction diagnosis in a large number of studies [40], and other methods such as the reward-based eating scale [41] or using nausea and increased salivary cortisol elicited by naltrexone as a marker of food addiction [42].

There are several similarities between addicted and obese persons: obese people and drug-addicted individuals [43] both tend to be more heavily influenced by immediate rewards and less responsive to future consequences. Moreover, impulsivity and inhibitory control are impaired in both overeating and drug addiction [44]. Obese individuals have the same enhanced attentional processing of food and food-related stimuli as individuals with addictions to cocaine, opiates, alcohol, nicotine, cannabis, and caffeine have with drug-related stimuli [45]. In addition, people with a higher BMI score higher on neuroticism and extraversion and lower on conscientiousness, personality traits typical of drug-addicted individuals [46].

However, the possibility that food may be as addictive as drugs is still largely debated since feeding, unlike drug self-administration, is a natural reward necessary to survival [10,47]. As a consequence, to date, food addiction has not been listed in the Diagnostic and Statistical Manual of Mental Disorders (DMS V) [48], and the concept still appears controversial. Moreover, obesity is a heterogeneous and multifactorial pathology that can depend on lifestyle or on other hormonal, genetic conditions, which is different from drug addiction [45,49]. Unlike addictive drugs, it is difficult to distinguish between normal consumption and compulsive abuse. Furthermore, the addictive component of food has yet to be clarified [49]. Some evidence suggests that combinations of macronutrients are an important determinant of compulsive overeating: modern, highly processed foods rich in fat and refined sugars are involved in problematic addictive-like behaviors [50].

In addition to defining and measuring food addiction, another important focus of research has been investigating the mechanisms underlying it. What are the neural circuits involved in hedonic eating, and what are the variations that need to occur to cause addiction? First of all, it is important to acknowledge that eating is intrinsically rewarding and that the consumption of palatable food activates the reward system. The neural circuitry systems responsible for regulating homeostatic and hedonic feeding are heavily integrated and regulate feeding behavior interdependently [51,52]. They usually harmonize their actions well because, in a physiological state, metabolic signals and the pleasure of eating are coordinated to drive food consumption [34]. Foods high in fat and sugar impair both the metabolic and the reward system; the development of metabolic or hedonic obesity is linked to different sets of susceptibility genes [34,53]. In people addicted to food, overeating may not cause leptin resistance, for example, but may cause changes in the reward system, leading to the uncontrollable desire for food and excessive food intake [53,54]. In accordance with this hypothesis, obese people show higher responsivity in reward and attention regions of the brain in response to food cues [55].

As with any other addiction, food addiction is characterized by three salient phases, which allow us to distinguish occasional and controlled use from chronic dependence: binge/intoxication, withdrawal/negative affect, and preoccupation/anticipation (craving) [55]. It is relevant to note that both the neurochemical substrates affected by rewarding foods (see Figure 1) and the symptomatology described by overeating humans in controlled trials overlap with the major findings described for drug addiction (for review, see [56]). Thus, food, especially palatable meals, activate the dopaminergic neurons of the ventral tegmental area (VTA), leading primarily to positive reinforcement and promoting food-seeking behavior consolidation. In this regard, it is important to note that the dopaminergic cells from the VTA are modulated by multiple signals (neurotransmitters, hormones, nutrients, immune signals), many of them coming from the periphery, making the VTA the main hub for reward regulation. Repeated exposure to palatable foods results in a decreased dopaminergic response and a reward deficiency that triggers food cravings and overeating. In parallel, incentive salience of food-associated stimuli rises, leading to contextual conditioning and Pavlovian instrumental transfer, which reinforces impulsive eating at the early stages.

After repeated cycles of food bingeing, the orbitofrontal cortex/prefrontal cortex shows a loss of executive control, eventually resulting in compulsive overeating. This transition is accompanied by affective allostatic changes, leading to a real food withdrawal syndrome, similar to that observed with major drugs. Thus, food withdrawal is associated with a negative emotional component that ultimately reinforces compulsive food intake. All of these elements have been characterized in animal models and observed in humans, but despite this solid evidence, the literature still lacks a consensus on the need to include food addiction as a disease. Only pica, rumination disorder, avoidant/restrictive food intake disorder, anorexia nervosa, bulimia nervosa, and binge-eating disorder are included in the DSM-V manual in its section on eating disorders, as stated before.

As shown in Figure 1, the bingeing stage involves VTA dopaminergic neurons projecting to nucleus accumbens (ACC) and dorsal striatum (DLS); the withdrawal stage is due to the activation of the amygdala (AMY), stria terminalis, ACC, and the subsequent recruitment of hypothalamic nuclei and brainstem autonomic circuits controlling stress responses. The craving stage is due to the prefrontal cortex (PFC), hippocampus, and insula activation. Compulsive behaviors and loss of control rely on the loss of orbitofrontal cortex projection to the dorsal striatum, which causes the emergence of habit-like behaviors [55]. In all these areas, the main impairments causing food addiction involve the dopaminergic, opioid, and cannabinoid systems, which are strictly interconnected. Their role has been confirmed by the preclinical and clinical results of different psychoactive medications (pharmacologically acting on these systems) in changing food consumption [55]. In particular, the opioid and cannabinoid systems are associated with pleasure derived from food and modulate functions of palatability (“liking” of food [34]). They also act synergically with dopamine (DA) in promoting food intake [57] (“wanting” of food), causing an increased release of DA in response to increased glucose and insulin [37]. In the case of the ECS, it is important to note that it is a major regulator of gamma-aminobutyric acid (GABA) and glutamate release through its retrograde messenger role in synaptic transmission, being a major substrate for adaptive learning associated with behavioral changes linked to addiction [58,59]. Moreover, some studies have underlined that the ECS could be a major regulator of bioenergetics in the brain, as revealed by its role as a modulator of neuronal and glial mitochondrial respiration, with important consequences for learning and memory [60,61]. Thus, the ECS position in the brain is at the core of the molecular adaptions associated with addiction, and understanding its role might help understand how the loss of control over eating influences the development of obesity.

## 3. The Endocannabinoid System

It is important to understand the particular characteristics of the ECS. It is an on-demand signaling system made up of endocannabinoid receptors, endocannabinoid signaling lipids, and the enzymes for their synthesis and degradation. Specifically, endocannabinoid receptor 1 (CB1) was first isolated in the central nervous system (CNS), where it is mainly expressed as one of the most abundant G protein-coupled receptors. In rodent brains, CB1 is mostly expressed in axon terminals from projecting neurons of cortical glutamatergic cells, on GABA terminals in basal ganglia, extended AMY, substantia nigra, globus pallidum, cerebellum, hippocampus, and the hypothalamus, where it regulates appetite, learning, memory, mood, and reward [62]. Principally, CB1 receptors are found presynaptically, where they inhibit neurotransmitter release after activation by endocannabinoids produced post-synaptically. As opposed to CB1, cannabinoid receptor 2 (CB2) was thought to be expressed only in immune cells. However, in the last few years, CB2 expression was also studied in the CNS, particularly in microglial cells. It was determined that it is, in fact, part of a general neural protective system modulated by homeostatic signals, stress, infection, or insults to brain parenchyma [63].

Both CB1 and CB2 are coupled to a Gi/0 protein, so their activation decreases the production of cyclic AMP (cAMP) and inhibits protein kinase A (PKA), stimulates the mitogen-activated protein kinase (MAPK), P38, Rho kinase, ROCK affecting synaptic plasticity, cell migration, and neuronal growth. Both receptors can be phosphorylated by G protein receptor kinases (GRKs) and associate with β-arrestin, leading to receptor internalization and extracellular-signal related kinase (ERK) pathway activation [64]. A special location that has lately been evidenced for CB1 receptors is the mitochondrial external membrane; activation of these mitochondrial CB1 receptors reduces respiration and ATP production, also leading to a decrease of presynaptic activity [60]. Figure 2 shows pathways of CB1 and CB2 activation.

CB1 and CB2 receptors are activated by endocannabinoids 2-arachidonoilglicerol (2-AG) and anandamide (AEA). Differing from other neurotransmitters, these molecules are small lipidic signals synthesized “on-demand” and are not stored in lipid vesicles in neurons. Moreover, they act mostly presynaptically rather than postsynaptically [65]. Other receptors that are suggested to mediate endocannabinoid action include the orphan G receptor GPR55 and the transient vanilloid receptor type 1 TRPV1 since 2-AG and AEA showed an affinity for both of those receptors [65].

Essentially, 2-AG synthesis takes place from 2-arachidonoyl-containing diacylglycerols (DAG) due to enzymes diacylglycerol lipase (DAGL) α and β. Several studies demonstrated that DAGLα is the most abundant isoform in the brain, whereas DAGLβ expression was reported in the same tissue, but it is more abundant in the periphery. DAG precursors come from hydrolysis of membrane phospholipids on account of protein lipase C (PLC) or from phosphatidic acid hydrolysis [65]. Additionally, 2-AG degradation occurs due to the enzyme monoacylglycerol lipase (MAGL), situated in synaptic terminals. In certain conditions, 2-AG can also be oxidated by cyclooxygenase 2 (COX2), generating prostanoids, which are important mediators of inflammation [66]. Figure 3 shows 2-AG synthetic and degradation pathways.

Brain 2-AG levels are 170 times higher than those of AEA, and the activation of the CB1 and CB2 receptors by 2-AG is associated with many physiological processes, including inflammation, food intake, locomotor activity, learning and memory, epileptogenesis, neuroprotection, pain sensation, mood, stress and anxiety, and addiction and reward [67].

Differently, AEA synthesis occurs in three different ways:Directly, from the hydrolysis of N-arachidonoyl-phosphatidyl ethanolamine (NArPE), a phospholipid precursor belonging to a complex family of lipids, the N-acylethanolamides (NAEs), through the enzymatic action of N-acyl phosphatidylethanolamine phospholipase D or NAPE-PLD;Through NArPE deacetylation by α/β-hydrolase domain type-4 (ABHD4) and the hydrolysis of glycerophosphoethanolamine by glycerophosphodiesterase GDE1;Via the PLC-mediated hydrolysis of NArPE to yield phosphoanandamide, which is dephosphorylated to AEA by a phosphatase.

The most common pathway is described first [65]. Several studies indicated that AEA exerts an overall modulatory effect on the reward circuitry [68], neuroinflammation, microglial activation, central control of blood pressure, and pain modulation [69]. Interestingly, hydrolysis of NArPEs can generate non-cannabinoid NAEs, such as oleoyl ethanolamide (OEA), capable of counteracting AEA effects on eating and metabolism [70]. AEA degradation happens due to enzyme fatty acid amide hydrolase or FAAH [66]. Inhibition of FAAH is able to increase the endogenous tone of NAEs [71], with important therapeutical and experimental implications. Additionally, AEA can be metabolized by COX2 giving prostanoids, or, alternatively, it can be hydrolyzed by N-acylethanolamine-hydrolyzing acid amidase (NAAA) [66]. Differing from 2-AG, AEA is only a partial agonist of CB1 receptors [62]. Figure 4 shows AEA synthetic and degradation pathways.

As mentioned above, the same mechanism that generates AEA might release other important lipidic mediators, which do not show particular affinity towards CB1 and CB2 receptors, but that share AEA biosynthetic and degradation pathways. Of these, it is worth mentioning palmitoylethanolamide (PEA) and OEA.

PEA is produced both in the periphery and in the CNS and has anti-inflammatory and pain-modulating effects [65]. It also has analgesic, antiepileptic and neuroprotective actions [62]. PEA actions have been linked to interaction with GPR55, peroxisome proliferator-activated receptor alpha (PPARα), and/or CB2 receptors [63]. Like PEA, OEA is produced in the intestines and in the CNS. It is known for its hypophagic [72] and neuroprotective effect [73] due to its binding with PPARα and TRPV1 receptors [72]. Interestingly, OEA seems to be an essential signal for regulating the response of the reward system to fat intake [74].

As can be deduced from the above description, the regulatory actions of cannabinoid receptors on neural transmission and glial function explain the multiple roles in which the ECS is involved: learning, memory and cognition, motor activity, mood tone, appetite and food intake, reward and addiction, neuroprotection, neural development, and sleep [75]. Referring in particular to food intake, the ECS is strongly implicated in the central control of energy homeostasis. In addition to central effects, the ECS also regulates eating and energy homeostasis acting at peripheral targets that include the gut, the liver, the adipose tissue, the endocrine, pancreas, and the peripheral sensory terminals [76].

Endocannabinoids facilitate the consumption of palatable food and fat accumulation, but at the same time, they also help to conserve energy in conditions of food scarcity [77]. As a matter of fact, fasting increases endocannabinoids levels in forebrain limbic structures [78]. In the forebrain, particularly in the cortex, glutamatergic neurons expressing CB1 promote eating by increasing odor processing and activating the sympathetic activity [77], whereas in the DLS, GABAergic neurons expressing CB1 modulate anorexigenic effects [79]. Moreover, endocannabinoids regulate the mesolimbic DA system, reinforcing the rewarding effects of food. Thus, fasting increases endocannabinoids concentrations within the ACC, activating the release of DA and driving liking and motivation to consume palatable foods [80]. CB1 mediates hypophagia or hyperphagia in ACC, activating respectively, GABAergic or glutamatergic neurons. On the other hand, cannabinoids mediate insulin-induced long-term depression in VTA, which decreases DA release within the mesocorticolimbic system [81], with anorexigenic effects.

Of course, the ECS is also part of all the integration signals in the hypothalamus that control food intake. Variations in the diet or nutritional status can vary endocannabinoids concentrations within this area of the brain [81]. In the paraventricular nucleus, CB1 receptors modulate the release of serotonin (5-HT) and GABA to stimulate food intake [82], whereas, in the arcuate nucleus, CB1 receptors are expressed in pro-opiomelanocortin (POMC) neurons, where their stimulation causes the release of the orexigenic peptide β-endorphin and the inhibition of the release of anorexigenic peptide α-MSH [83]. Several interactions between leptin, ghrelin, orexin, and the ECS in the hypothalamus have been widely studied and reviewed by Lau et al. [81]. Further investigation is needed to understand the interplay between the mesolimbic DA system and the hypothalamus. Both homeostatic and hedonic eating depends on the interactions between endocannabinoids, DA, leptin, orexin, and ghrelin as explicated in [84], taking into account that in the lateral hypothalamus, an area rich in CB1 receptor expression, there are GABAergic neurons projecting to DA ones in VTA, integrating in this way homeostatic food intake with reward aspects [84].

Additionally, as mentioned above, the ECS has a strong role in the gut-brain axis; cannabinoid receptors are expressed on peripheral terminals of sensory neurons, in parasympathetic and sympathetic terminals, and on vagal afferences and efferences regulating gastric motility [85]. Production of endocannabinoids is regulated by fasting and refeeding and modulates appetite and gut hormone secretion peripherally [76].

## 4. Endocannabinoid System Physiopathology in Obesity and Food Addiction

Pathological alterations to the ECS have been described in nearly all chronic disorders and, in particular, addiction, neurodegenerative diseases, mood disorders, and obesity [65]. Obesity has been linked to an overactive ECS, as evidenced extensively in the literature, for example [81,86,87,88]. Plasma endocannabinoids correlate positively with markers of obesity and metabolic disorders [89,90], and alterations to the genes encoding CB1, CB2, and FAAH have been associated, respectively, with metabolic syndrome and overweight and obesity [85,91]. However, rodents displaying these genetic variations show increased body weight and metabolic impairments only when fed with a high-fat diet and not chow [92]. Both people carrying these polymorphisms and obese individuals present higher levels of circulating cannabinoids and show greater reward-related striatal activity [53,85,89].

It is important to underline that the changes in plasma cannabinoids may be influenced by eating. Western diets rich in refined food contain higher levels of polyunsaturated fatty acids (PUFAs), whose increased intake can influence the production of both AEA and 2-AG since they are derivatives of PUFAs [84,93]. Moreover, increased plasma cannabinoids may also derive from disrupted lipid metabolism (a chronic condition common in obesity [94]) or by genetic alterations to proteins linked with energy balance and fat metabolism (for an example, see [95]). The demonstrated changes in plasma cannabinoids are relevant when discussing obesity and food addiction since recent data demonstrated that blood levels of these molecules might influence taste sensibility and, consequently, food intake [96,97].

With Rimonabant studies in obesity (RIO), researchers started to focus on the role of the ECS in the periphery and, in particular, in metabolism [76,98]. Peripherally, various variations to the ECS have been observed in obesity. In adipocytes, the pathology causes increased endocannabinoid tone and CB1 expression and a decrease in NAPE-PLD, promoting fat mass accumulation. In the liver, exposure to a high-fat diet increases AEA but not 2-AG levels as a result of decreased FAAH expression, whereas in the muscle, fat intake causes reduced CB1 expression [77].

Although this fat accumulation, which promotes endocannabinoid overactivation, is relevant for “metabolic obesity”, it does not explain the role of this system in food addiction. Thus, to serve the aim of this review, it is important to move our focus again into the brain. Little has been acknowledged or published in the literature about variations that adversely affect the ECS in the CNS in obesity. In particular, it is known that the obesity pathology is associated with increased: (a) endocannabinoid tone in the brain [90] and( b) increased CB1 expression [89]; moreover, it has been considered (c) responsible for the reinforced motivation to consume highly palatable foods [99].

Regarding the obese phenotype, studies have focused on the main regulatory center, the hypothalamus, and several brain areas connected to it.

*Hypothalamus.* Some studies focusing on the hypothalamus have demonstrated that increased endocannabinoid tone in this region is responsible for synaptic remodeling of excitatory and inhibitory synapses, causing the inhibition of anorexigenic neurons and, consequently, hyperphagia and weight gain [99]. Diet-mediated endocannabinoid changes in this region have also been associated with impaired thermogenesis and leptin response activation as a physiological compensatory response [100]. However, recent studies demonstrated that both enhanced endocannabinoid tone and increased CB1 expression in the hypothalamus are transitory and characteristic only of the early phases of exposure to HFD [100]. In the same area, other studies displayed altered expression of CB2 [101], NAPE-PLD, and DAGLa [102]. Additionally, significant gender differences were found in these results [100,102], suggesting the existence of sex-specific variations.

*Other brain areas.* ECS alterations related to obesity have also been shown in the arcuate nucleus, with a significant increase in 2-AG and CB1 mRNA and protein expression in response to diet-induced obesity in rats [103]. Increased CB1 expression was also reported in the nodose ganglia and nucleus of the solitary tract of Zucker and Sprague Dawley rats fed with a high-fat diet, where the mechanosensitivity of gastric vagal afferent neurons are controlled [104]. On the other hand, diet-induced obesity was related to a lower density of autoradiography CB1 expression in the hippocampus, cortex, and nucleus accumbens [105]. Decreased CB1 mRNA expression after exposure to high-fat diets was observed in the limbic forebrain [105] and VTA [106], while a significant decrease for the expression of all endocannabinoid enzymes was displayed in VTA and ACC. At the same time, in both areas, a significant increase in the expression of the CB2 receptor was shown [106]. Moreover, in VTA, unlimited access to a high-fat diet increased 2-AG levels and abolished insulin-induced long-term depression [107].

Studies addressing the role of the ECS in food addiction have been more complex because of the lack of real models of compulsive eating. Thus, comparatively few studies have investigated the effects of exposure to a palatable diet in humans or rodents in this signaling system. What is widely known is that the ECS regulates the palatability of foods [108] since receptors mediating taste coexpress with CB1 receptors in the circumvallate and foliate taste bud cells [109]. That palatability displays plasticity since flavor preferences are acquired and change throughout life [110]. ECS is known to enhance “liking” [111] and ease “wanting” foods by facilitating the activation of mesolimbic DA neurons [112]. Both effects are principally mediated by CB1 receptor activity [113]. In fact, cannabinoids are involved in all hedonic processes since they can affect the brain reward system and reward-related behaviors [114] by enhancing DA transmission in ACC and DA neuron firing in VTA [115,116].

Several mechanisms involving ECS may be at the base of food addiction. Coccurello et al., in their review [84], suggested that food addiction-like behavior could be caused by impaired interaction between orexin/orexin1 receptor and 2-AG/CB1 in the lateral hypothalamus (see the position of these neurons in Figure 1), ultimately leading to VTA DA neuron activation, which induces relapse to food consumption, similarly to what happens with cocaine [117]. These types of interactions occur not only during positive reinforcement but also during tolerance and abstinence, leading to negative effects after prolonged exposure to palatable foods (reward deficits). Abstinence from high palatable food causes a withdrawal state similar to that associated with drugs of abuse, which may result from ECS alterations in AMY [118,119]. Other mechanisms involved in food addiction refer to CB1 expressing glutamatergic neurons in the prelimbic cortex projecting to ACC. Selective glu-CB1-knockout mice displayed resilience to food addiction [120], whereas selective silencing of these neurons resulted in compulsive eating behavior, with animals unable to stop palatable food consumption despite negative consequences [120]. In fact, addictive behavior is associated with alterations in synaptic transmission of prefrontal areas that normally mediate inhibitory control.

Moreover, preclinical studies have demonstrated that limbic levels of endocannabinoids correlate with cravings for tasty food [121] and that CB1 antagonism by Rimonabant administration was able to decrease (in a dose-dependent manner) the preference for palatable food [122], preventing the increase of DA release in ACC [116]. Rimonabant was also able to reduce excessive overeating and risk-taking behavior of compulsive eating in a high-sucrose sweet-fat diet model of rats [123]. Consequently, CB1 agonists and antagonists dose-dependently increased or reduced, respectively, motivation for food in food self-administrating rats [124]. On the other hand, administration of CB1 agonists induced food intake also in satiated animals by activation of the corticostriatal-hypothalamic pathway [125], while administration of tetrahydrocannabinol (THC, CB1 agonist) was able to increase DA release in ACC in response to sucrose administration [126]. Other evidence supporting the role of the endocannabinoid system in hedonic feeding is the fact that CB1 knockout mice displayed reduced motivation to work for food [127] and presented a lack of impulsivity when compared to wild-type mice trained with chocolate-flavored pellets [128]. Finally, a relevant role for the cannabinoid CB1 receptors in food preference was discovered in the olfactory system [129], where they regulate food odor detection by modulating the cortical glutamatergic input into the olfactory bulb. To what extent odor detection contributes to food addiction in humans is still unknown, although the chemosensory properties of foods are essential for the rewarding process associated with their ingestion through the modulation of ghrelin, a CB1 receptor-modulated hormone [130].

CB1 appears to be the most important element of the ECS in addiction, but limited evidence demonstrates that the CB2 receptor may also be involved. CB2 blockade or genetic deletion inhibits dose-dependent reduction in sucrose self-administration caused by systemic administration of cannabidiol (CBD) [131].

In addition to CB1 and CB2 receptors, clinical and preclinical studies have suggested that people and animals presenting specific FAAH polymorphisms (thus altering the rate of degradation of AEA and related NAEs) not only show an obese phenotype but also display higher D3 DA receptors’ expression in the brain. That exposes these subjects to greater addiction risk [132]. Similarly, FAAH inhibition stimulates feeding with a high sucrose/high-fat diet, thus sustaining a role for this enzyme in regulating hedonic feeding behavior [133,134].

Other evidence linked to ECS involvement in food addiction is that cafeteria-style diets (rich in sucrose and fat) are able to dysregulate the expression of cannabinoid receptors and cannabinoid biosynthetic and degradative enzymes [131].

Regarding the endocannabinoid lipids, there are pharmacological studies supporting an endocannabinoid role in food addiction; injections of AEA in ACC enhanced food palatability [80].

Few studies have focused on food addiction in humans, but clinical studies by Monteleone [135] demonstrated altered plasma levels of cannabinoids in patients suffering from eating disorders such as anorexia nervosa and binge-eating. Most interestingly, Monteleone discovered that 2-AG and AEA levels in plasma increased before and after palatable food consumption [136] and that, conversely, 2-AG levels decreased in plasma of obese people consuming non-palatable food [137]. All these data suggest a role for 2-AG in anticipation and reward since it can trigger preference towards the consumption of palatable food [138]. As a matter of fact, the administration of physiological doses of 2-AG and AEA increased human taste cell response to sweeteners by more than 120% [109].

## 5. Endocannabinoid System as a Pharmacological Target for Obesity and Food Addiction

All the studies cited above have shown a pivotal role for central and peripheral ECS in obesity and on the regulation of food intake, which makes it a possible pharmacological target to treat these disorders. Firstly, all the attention was focused on CB1 antagonists, as the CB1 receptor seems to be the most implied in the control of energy homeostasis, both at central and peripheral levels. All this interest culminated in the FDA and EMA approval of the anti-obesogenic drug **Rimonabant (Sanofi)**, which, unfortunately, was retired after a short time due to important adverse effects observed after chronic administration to obese people. These effects were mainly psychiatric and consisted of anxiety, depression, and suicidal ideation [139]. Negative emotional states caused by Rimonabant were attributed to its action on central CB1 [140], and recent research discovered that systemic administration of the drug was able to selectively increase 2-AG and CB1 mRNA transcription in central AMY, suggesting that this brain region may have a key role in the displayed anxiety and depressive-like behaviors [141]. It is of special relevance that all the CB1 receptor antagonists tested in clinics were, in fact, inverse agonists, a class of drugs that are capable not only of blocking but actually reversing the endocannabinoid signal at those receptors. Neutral antagonists or weak inverse agonist properties might have resulted in a different outcome of the clinical trials, but that was never addressed.

Since then, anti-obesogenic treatments have focused on other possible pathways regulating the pathology that culminated in FDA approval of the 5-HT2c receptor agonist lorcaserin (Belviq), glucagon-like peptide-1 (GLP-1) receptor agonist liraglutide (Saxenda), or the complex polypharmacological combination of phentermine/topiramate in extended-release (Qsymia) or naltrexone/bupropion (Contrave) [142]. Between all these available treatments, only liraglutide and naltrexone/bupropion combinations have been approved as weight loss agents by EMA [143]. However, all these treatments present important side effects. No one-size-fits-all treatment has been found to date [144], perhaps because drugs must not only promote weight loss and body fat reduction, but they also have to change the way the brain responds to food [55]. In addition to these data, studies have shown interactions between endocannabinoid-like molecules and GLP-1 [145]. Preclinical results demonstrated that CB1 antagonism could enhance GLP-1 receptor agonists activity [146] and improve GLP-1R mediated insulin secretion [147], suggesting that new drugs acting on the ECS system may have a synergic effect and ameliorate the results obtained with liraglutide treatment. For these reasons, research on anti-obesogenic drugs acting on the ECS is still an option that must not be discarded. Murphy et al., in their review, summarized the effects of several drugs acting on the ECS in preclinical and clinical studies on body weight over the last few years. Unfortunately, none of these molecules were good enough to gain FDA or EMA approval for obesity treatment [148].

Centering our attention on CB1-acting molecules, after Rimonabant’s withdrawal, the first studies were focused on its analog, **Taranabant**. Clinical and preclinical investigations on this CB1 inverse agonist were abandoned due to the adverse reactions experienced with Rimonabant, considering its similar pharmacotherapy [149]. For the same reason, no clinical study to date has been run for promising CB1 inverse agonist **AM251** [148,150], despite several preclinical studies demonstrating its capability to reduce body weight, increase energy expenditure and induce hypophagia [151].

Central cannabinoid CB1 receptors play a pivotal role in controlling food intake and reward-associated stimuli, but adverse effects of anxiety and depression deriving from its modulation in the brain precluded its use in clinical practice; therefore, research focused on the role of peripheral cannabinoid receptors as modulators of appetite. In fact, peripheral actions on ECS and consequential variations to the metabolism result in the modulation of several peptides and hormones related to energy balance. These could also have a central action (for review, see [152]). In addition, recent discoveries demonstrated that genetic ablation of cannabinoid CB1 receptors only in the adipose tissue caused marked changes in brain function and plasticity, a clear demonstration of the bidirectional dialogue between central and peripheral tissues in the regulation of weight and energy expenditure homeostasis through the ECS [153]. The possibility of having anti-obesogenic effects by using peripheral restricted neutral cannabinoid CB1 receptor antagonism was attempted using the poor brain penetrant drug **LH-21** [154,155]. This drug was able to prevent the onset of type 2 diabetes and obesity-induced hypertension [156]. This proof of concept was followed by the development of other peripherally restricted CB1 inverse agonists. The most promising molecules of this group were **TM38837** [157], **AJ5012** [158], **AJ5018** [159], **JD5037** [160], **BPR0912** [161], **BPR697** [162], **TXX-522** [163], **ENP11** [164], **Compound6a** [165], **BMS-725519, BMS-811064,** and **BMS-812204** [166]. All these molecules showed a significant reduction of body weight, food intake, or metabolic activity and were characterized by a low brain/plasma concentration ratio. That meant that the anxiety-like side effects were only observed at dosages several times higher than the recommended values [148,165]. Even though preclinical studies assured the efficacy and safety of these drugs, only TM38837 has been involved in nonhuman primate studies to investigate its low blood-brain barrier permeability. It was also used in a single ascending dose study in healthy male volunteers (who did not show psychiatric side effects [167]). On the other hand, JD5037 was approved by the FDA as an investigational new drug [165]. Studies proved its preclinical efficacy for Prader–Willi syndrome and renal impairment induced by diabetes [168], but stereotypical neurobehaviors were observed in male and female rats after its administration [169]. Another CB1 inverse agonist compound (which still calls for investigation but has shown promising results for the treatment of obesity while maintaining low plasma/brain concentration) is **Compound2p**, whose administration showed enhanced glucose absorption in obese animals [170].

Surprisingly, some CB1 agonists have shown weight loss promotion and protection against obesity, perhaps acting as functional antagonists in vivo, competing with endogenous 2-AG [171]. The most studied CB1 agonist has been **WIN 55212** [122], whose administration has been associated with decreased food intake and slowed weight gain. These are effects that can be potentiated by synergic administration of exendin-4 (glucagon-like peptide-1 antagonist). Most importantly, CB1 agonists such as WIN 55212 do not cause psychiatric effects and have been studied for their ability to alleviate chronic mild stress [148]. Other CB1 agonist compounds investigated for their anti-obesogenic activity include **CP-55940** [172] and **CBDD** [173]. Although some of these molecules have reached phase 2 of clinical trials, none have been approved for treatment [174], considering their ability to exert central adverse effects.

Of course, CB1 partial central-acting agonist **THC** has been studied for its effects on food intake and body weight. In particular, THC treatment was unable to affect body weight and food intake in lean mice but was significantly able to reduce body weight, fat mass, and energy intake in diet-induced obesity mice [175].

Additional drugs acting on CB1 receptors include **AM6545**, a CB1 neutral antagonist with a 10-20% brain/plasma ratio, able to reduce food intake and body weight and ameliorate hepatic steatosis, insulin sensitivity, and dyslipidemia in preclinical animal models [176,177]. One more eligible candidate from this class is **AM4113** [150], which was proven effective in reducing the body weight of rats without inducing any symptoms of anxiety and depression despite its ability to cross the blood-brain barrier [148]. **THCV** is also part of this class; the CB1 neutral antagonist demonstrated increased energy expenditure in rats and decreased BMI-related connectivity between amygdala and precuneus in humans, suggesting a central mechanism behind its ability to modulate food intake [178,179]. An important factor supporting THCV use in therapy could also be its antipsychotic effect [180]. Other compounds with same pharmacodynamics, i.e., able to influence feeding behaviour, which still require further research include: **NESS06SM** [181], **SM-11** [182], **PIMSR** [183], **BAR-1** [184], **Compound10q** [185], **Compound2c** [186], **Compound5** [172], **Compound13** [165], **Compound6a** [187], and **CompoundD4** [188]. These were designed to have low penetration in the brain, except for Compound6a, whose chronic administration caused accumulation in the brain [187]. On the other hand, clinical studies demonstrated decreased body weight and a safe administration profile for two other CB1 antagonists: **AVE-1625** [174] and **SLV-319** [189].

In 2018, Amgen published an EP patent for a CB1 receptor antibody that claimed to cover all the diseases involving antagonism of CB1 receptors. An antibody for CB1 is in phase one trials for obesity, but still, no results have been published at this time [190].

In addition to all the molecules cited thus far, which act only on CB1 receptors, researchers have developed hybrid molecules targeting more than one receptor, aiming to reduce the adverse effects and potentiate metabolic effects. An example of these compounds is **MRI-1569,** a hybrid CB1 inverse agonist/inducible nitric oxide synthetase (iNOS) inhibitor that reduced food intake and body weight and improved glucose tolerance and hepatic steatosis [191]. The same molecule was effective in reducing voluntary ethanol intake in mice, suggesting it may be a good candidate to treat food addiction [192]. Other drugs act on both PPARα and CB1 receptors to exert their anti-obesogenic activity, as is the case for **Compound7** [193] and **OLHHA** [194]. For certain, the most numerous classes are made up of drugs acting both on CB1 and CB2 receptors. A notorious example is **URB447**, a hybrid CB1 antagonist/CB2 agonist able to reduce food intake and weight gain without exerting central effects [195]. Another molecule pertaining to this class, with the same pharmacodynamic profile as URB447 but with central activity, is **CBD**, able to significantly inhibit weight gain in high-fat diet-fed rats [196]. Additionally, some observational studies have investigated the possibility of using THC/CBD combination drugs as a viable alternative for obesity reduction [171]. Another compound that had promising results in reducing food intake is RVD-hemopressin(α), an allosteric modulator of both CB1 and CB2 receptors, able to act in the hypothalamus reducing POMC, Agouti-related peptide mRNA of cafeteria-diet-fed rats [197]. Another compound with double activity on CB1 and CB2 receptors is (+)-trans-Cannabidiol-2-hydroxy pentyl, whose administration in mice was able to prevent streptozocin-induced hyperglycemia and diabetic nephropathy. The effects of this compound on food intake or body weight have not yet been studied [198].

Of course, drugs acting on other components of the ECS have been developed, as well. In particular, the potential of CB2 receptor agonists as anti-obesity treatments has been of interest since CB2 receptors are highly expressed in the brain and regulate glucose and lipid metabolism [174]. All molecules developed until now act at both the peripheral and central levels, reducing the inflammatory state related to obesity and increasing excitatory synaptic transmission without eliciting psychotropic activity [199]. In particular, molecule **JWH-105** was able to decrease body weight in obese mice, whereas **JWH-133** reduced transcription of inflammatory biomarkers in human adipose tissue [174]. Additionally, a hybrid molecule acting on peroxisome proliferator-activated receptor gamma (PPARγ) and CB2 receptors was studied as a potential treatment for obesity. **VCE 004.8** administration in rats induced decreased body weight, total fat mass, adipocyte volume, plasma triglyceride levels, leptin levels, ameliorated glucose tolerance, and increased adiponectin and incretin levels [200].

However, other pharmacological strategies focused on regulating endocannabinoids tone by acting on the enzymes of synthesis and degradation. There is no current evidence published on drugs modulating NAPE-PLD, FAAH, and MAGL expression with anti-obesity-like effects, even though several molecules have been developed to treat CNS-related pathologies. **O-7640**, a relatively selective inhibitor of the synthesis of 2-AG, has had promising preclinical results; it counteracted hyperphagia, reduced the intake of palatable food, the amounts of 2-AG in the hypothalamus and liver, and also reduced body weight because of its strong inhibitory action on DAGLα [201].

In any case, all the drugs cited thus far are not endogenous molecules. Other studies have investigated the role of endocannabinoid-like compounds already present in our bodies and their role in the modulation of food intake, both at central and peripheral levels. Examples include **PEA** and **OEA** [202], whose administration was able to decrease food intake [72,203], or **C18:1 NAT**, which mediated reversible changes in food intake [204]. However, while they still are structurally related to endocannabinoids and share biosynthetic and degradation pathways, these molecules do not interact with cannabinoid receptors, so they do not fall within the main goal of the present review.

Table 1 sums up all the cited drugs and their pharmacodynamics.

In summation, several drugs acting on the ECS have been or are currently being investigated as novel pharmaceutical treatments for obesity. The fact that the majority of these molecules are able to modulate food intake or act on energy expenditure makes them perfect candidates for the treatment of food addiction, whose primary component is hyperphagia [73]. Unfortunately, except for SM-11 [182], no molecule so far has been tested in a model of food addiction, and there is a notable lack of information about the capability of these molecules to act on the reward system. However, the ECS has a well-recognized role in addictions [101], and targeting it could be beneficial towards efforts to re-establish reward alterations related to disrupted eating behavior [9], even though further studies would be required to assess the role of the cited molecules in reward system impairments.

Future directions for the study of ECS targeted therapies in obesity should be more oriented to modulation of the enzymes of synthesis and degradation of cannabinoids (the study of which has been insufficient until now), and/or identifying the harmful central mechanisms causing the anxiety and depressive-like behavior that led to the withdrawal of Rimonabant and its analogs. In fact, despite the intent of synthesizing peripherally selective molecules, many drugs cited here can cross the blood-brain barrier and exert a central action, which appears to be fundamental for their hyperphagic action. Furthermore, this review underlined the importance of the ECS in mediating reward related to food stimuli. It is the authors’ opinion that no effective treatment could be achieved for obesity without normalizing reward system disruptions caused by food addiction. Thus, further investigation may be required on those molecules which act on the ECS, exert central effects, and have desirable brain/plasma ratios that would not cause anxiety and depression. Of course, clinical evaluation of these molecules must thoroughly investigate their effects on endocrine, immune, reproductive system, and emotional behavior, whose activities are strongly modulated by the ECS.

## 6. Conclusions

Obesity is a complex disorder, and the number of obese individuals is growing swiftly. During the last several years, research has focused on a hypothesis: obesity is a food addiction caused by reward impairments related to hedonic eating. The ECS plays an important role in the central control of food intake and reward-related behaviors and regulates peripheral tissues activity in the gastro-intestinal tract or adipose tissue that can modulate the brain reward system. Moreover, both obesity and food addiction have been linked to impairments to the ECS in various brain regions. For these reasons, targeting the ECS could be a valid pharmacological therapy for these pathologies. However, further studies are necessary to find safe and effective treatments that can act at both the peripheral and central levels, exerting anti-obesity effects and normalizing reward-related behaviors without causing harmful adverse effects.

## Figures and Tables

**Figure 1 pharmaceuticals-14-01002-f001:**
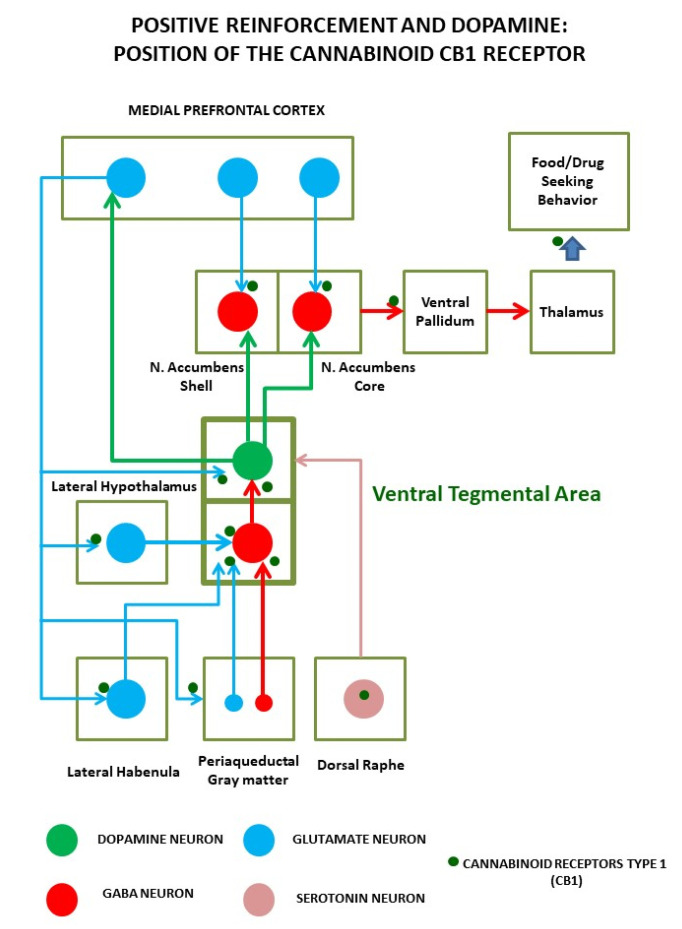
Positive reinforcement induced by palatable foods (and abused drugs) relies on the activation of a complex circuit around the dopaminergic neurons of the ventral tegmental area. These neurons not only detect ongoing reinforcements but code anticipation of future access to them. Dopamine cells are sensitive to multiple inputs, including hormones and nutrients, and are regulated by cortical, subcortical, and brainstem projections, mainly from gamma-aminobutyric acid (GABA) and glutamate cells. Precisely, in the axon terminals of these neuronal inputs, cannabinoid receptors regulate neural transmission and plasticity events that lead the transition from controlled intake to the compulsive intake that characterizes addiction.

**Figure 2 pharmaceuticals-14-01002-f002:**
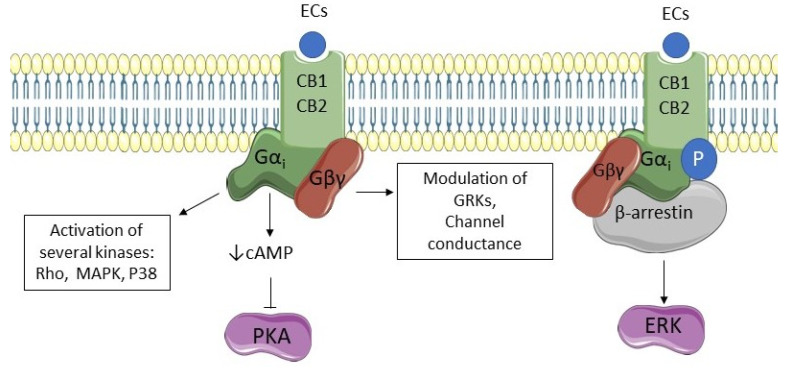
CB1 and CB2 signaling pathways. Endocannabinoids (ECs) stimulate cannabinoid receptors type 1 and 2 (CB1 and CB2, respectively), activating G-protein subunits. Subunit Gα decreases cyclic AMP (cAMP) and inhibits the activity of protein kinase A (PKA). Moreover, it is responsible for the activation of several kinases such as mitogen-activated protein kinase (MAPK), P38, and Rho kinase. Subunit Gβγ activation modulates channel conductance and the activity of G protein receptor kinases (GRKs). Both receptors can be phosphorylated by GRKs and associate with β-arrestin leading to receptor internalization and extracellular-signal related kinase (ERK) pathway activation.

**Figure 3 pharmaceuticals-14-01002-f003:**
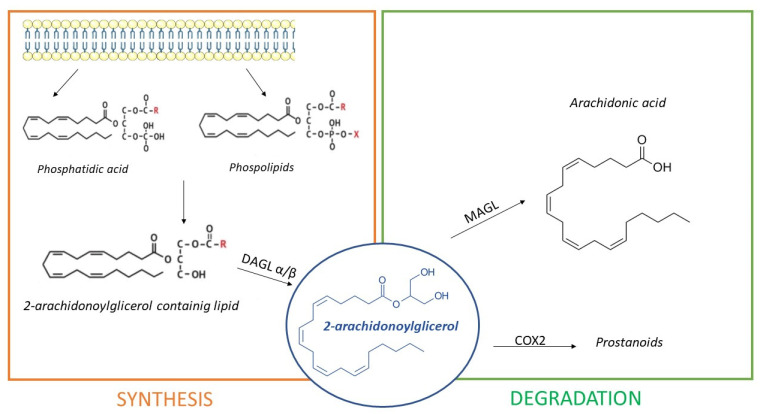
The synthesis of 2-Arachidonoylglicerol (2-AG). Synthesis happens because of diacylglycerol lipase (DAGL) α/β, from phosphatidic acids and phospholipids proceeding from the cell membrane. Additionally, 2-AG is degraded by monoacylglycerol lipase (MAGL) in arachidonic acid and by cyclooxygenase-2 (COX2) in prostanoids.

**Figure 4 pharmaceuticals-14-01002-f004:**
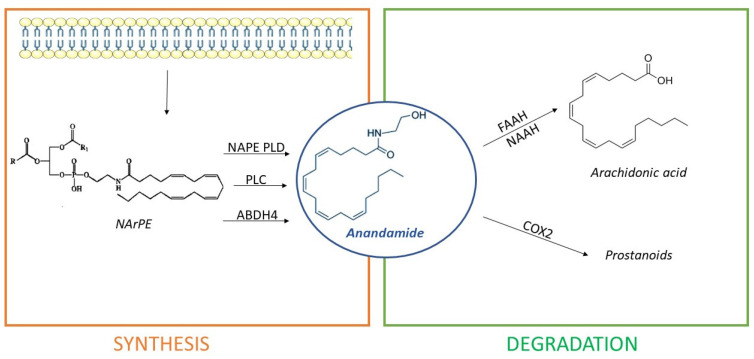
Anandamide (AEA) is synthesized from cell membrane lipid precursors through hydrolysis of N-arachidonoyl-phosphatidyl ethanolamine (NArPE) by N-acyl phosphatidylethanolamine phospholipase D (NAPE-PLD), α/β-hydrolase domain type-4 (ABHD4), or phospholipase C (PLC) action. It is then degraded in arachidonic acid by enzymes fatty acid amide hydrolase (FAAH) or N-acylethanolamine-hydrolyzing acid amidase (NAAA) or in prostanoids by cyclooxygenase-2 (COX2).

**Table 1 pharmaceuticals-14-01002-t001:** Anti-obesogenic drugs acting on the endocannabinoid system and their relative pharmacodynamics. Abbreviations used in the table: CB1; cannabinoid receptor type 1, CB2; cannabinoid receptor type 2, iNOS; inducible nitric oxide synthetase, PPARα; peroxisome proliferated-activated receptor alpha, PPARγ; peroxisome proliferated-activated receptor gamma, DAGL; diacylglycerol lipase.

Pharmacodynamics	Drug	Effects	References
Central CB1 inverse antagonists	Rimonabant, AM251	Reduced food intake, body weight gain, metabolic effects	[151,205]
	Taranabant	Increased energy expenditure, decreased caloric intake	[149]
Peripheral CB1 inverse antagonists	LH-21, TM38837, Compound 6a, BMS-811064, BMS-812204	Reduced food intake, body weight gain	[155,157,165,166]
	AJ5012, AJ5018	Reduced body weight gain, adipose tissue inflammation	[158,159]
	JD5037	Reduced food intake, body weight gain, metabolic effects	[160]
	BPR0912	Reduced body weight gain, thermogenesis modulation	[161]
	BPR697, TXX-522	Reduced body weight gain, metabolic effects	[162,163]
	ENP11	Reduced food intake, thermogenesis	[164]
	Compound 2p	Reduction in plasma glucose	[170]
CB1 agonists	WIN 55212, CP-55940	Reduced body weight gain	[122,172]
	CBDD	Metabolic effects	[173]
	THC	Reduce food intake, body weight gain, fat mass	[175]
CB1 neutral antagonists	AM6545	Metabolic effects	[176]
	AM4113, THCV	Reduced food intake, body weight gain	[150,178]
	NESS06SM	Reduced body weight gain, fat mass	[181]
	SM-11	Reduced food intake, self-administration of palatable food	[182]
	PIMSR	Reduced hepatic steatosis	[183]
	BAR-1	Reduced body weight gain, metabolic effects	[184]
	Compound10q, Compound2c, Compound5, Compound13, Compound6a, CompoundD4	Reduced food intake	[165,172,185,186,187,188]
	AVE-1625, SLV-319	Reduced body weight gain	[174,189]
CB1 antibodies	/	/	[190]
CB1 inverse agonist/iNOS inhibitor	MRI-1569	Reduced food intake, body weight gain, hepatic steatosis, metabolic effects	[191]
	MRI-1867	Attenuates obesity-induced chronic kidney disease	[191]
CB1/PPAR-α antagonists	Compound7	Metabolic effects	[193]
	OLHHA	Reduced food intake, metabolic effect, ameliorate non-alcoholic fatty liver disease	[194]
CB1 antagonist/CB2 agonist	URB-447, CBD	Reduced body weight gain	[195,196]
	RVD-hemopressin(α)	Reduced food intake, metabolic effects	[197]
CB2 agonists	JWH-105	Reduced body weight gain	[174]
	JWH-133	Reduced adipose tissue inflammation	[199]
PPAR-γ/ CB2 agonist	VCE 004.8	Reduced body weight gain, fat mass, metabolic effects	[200]
DAGL-α inhibitor	O-7640	Reduced food intake, body weight gain	[201]
Endocannabinoid-like molecules acting on other receptors	PEA, C18:1 NAT	Reduced food intake	[203,204]
	OEA	Reduced food intake, body weight gain, metabolic effects	[70]

## Data Availability

Reference to all data are contained within the article.

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
