# Peer review of "Obesity as a Condition Determined by Food Addiction: Should Brain Endocannabinoid System Alterations Be the Cause and Its Modulation the Solution?"

_pharmaceuticals, 2021, doi:10.3390/ph14101002_

Round 1

Reviewer 1 Report

The authors of the manuscript “Obesity as a condition determined by food addiction: should brain endocannabinoid system alterations be the cause and its modulation the solution?” have done a refreshing and brave review on a topic that sadly, within the field, has been set aside for the uncomfortable elephant in the room of Rimonabant’s fiasco. A very important point is brought into the subject regarding the possibilities of the ECS to modulate addictive behavior towards palatable food in westernized diets. The authors make an introduction about obesity, its complications, and its possible relation to food addiction. They further introduce the endocannabinoid system focusing on the central nervous system to later discuss its role in food addiction through its action in the brain. They also review centrally and peripherally acting synthetic cannabinoids and their anti-obesogenic properties. Although the text is joyful in the style, and the topic of importance, there are many points that need to be revised.

  • In general, the text would greatly improve if it would be more “storytelling” rather than “word listing”, and there is an overuse of the semicolon
  • Introduction: Obesity is introduced without mentioning at first why it is considered a chronic pathology. There is also an underevaluation of other factors very important in the development of obesity and overvaluation of food addiction; for instance, there is great relevance of what we eat rather than the density of calories, including process foods with certain types of fats and sugars, or also there is no mention regarding the importance of inflammation at all. As well there are many cases of what is known as “healthy obese” which are not mentioned in the introduction.
  • Authors have used primarily references of reviews instead of original work. References must be changed to include the original work that has been done, which sometimes is also just 1 or 2 references no more. When not feasible for references limitations, then state “reviewed in XX”, and try using the review from the people that described it
  • Authors are sometimes biased towards their field: they consider that very few studies describe the role of the ECS in feeding while they state that CB1 is in the mitochondria, although there are very few studies that have shown it.
  • Table 1: It is not possible to distinguish between different rows. It misses novel compounds with double activity over CB1 and CB2, for example (+)-trans-Cannabidiol-2-hydroxy pentyl.
  • The title of table 1 does not reflect its content: there is no mechanism of action in the table. It would, in fact, be more informative if another column is added to include the impact regarding food intake/addiction/obesity of the different targeting strategies mentioned in the first column.
  • Authors talk about palatability but do not mention that the receptors are expressed in the taste buds, regulating taste.
  • Line 456: authors do not really mention periphery before
  • Line 459: it was not known that CB1 antagonism could have an impact on metabolism through its action on peripheral tissues before rimonabant. Indeed, it was after the RIO study that benefits independently of food intake were noticed.
  • Regarding the compounds included in the text: the authors do not mention the specificity of the synthetic cannabinoids
  • To the knowledge of this reviewer, JD5037 is not moving into a further clinical trial. If authors have more information regarding that, please include a reference
  • Why do the authors talk about peripherally restricted compounds if the point of the review was to focus on the CNS and addiction? And what are the implications of the double-acting compounds (such as double CB1-iNOS) regarding food addiction?
  • Line 601: The ECS has been studied for the treatment of complications associated with obesity.
  • Line 602/603: not all of them neither the majority of them regulate appetite, nor the reward response. So they are not all perfect candidates to treat food addiction. There is overall a missing message: which compounds (to my understanding CNS penetrant?) would be good candidates for the treatment of food addiction, and which are the cons of using a brain-penetrant blocker of CB1? Do the authors expect a different outcome than Rimonabant? And why? This is very important from both the scientific and the pharmacological and clinical points of view. Authors must include the risks of this strategy and a risk management plan. Also, the results obtained (metabolically) by using synthetic cannabinoids that bypass the CNS in obesity are on many occasions very good. In the current review, the importance, if any, of targeting the CNS is missing in the message, and needs to be highlighted
  • The manuscript needs to be deeply proofread for typos and also checked by an English native/high-level speaker as it is full of mistakes, sometimes misleading the message. Examples:
    • Line 102, remove “that”
    • Line 104, wrong use of “currently”. Maybe “up to date”?
    • Line 115 fix sentence
    • Line 147, a category of what?
    • Line 201, wrong use of “Apparently”: that is something the authors heard on an elevator conversation but do not believe it. Either the authors are uncertain of what they write, so then remove the whole sentence as this is a review and not an elucidation, or it has a sarcastic meaning, which I do not believe it was the intention
    • In general, semicolons are used instead of what should be commas, full stops, verbs, prepositions, or nothing at all
    • Lines 217 and figure 2: GRKs instead of GRKS
    • Line 220 and figure 3 legend: 2-ARACHIDONOYLGLYCEROL!!!!! 2-Acylglycerol is something else
    • Also, ECs are not “lipid signals” they are lipids that signal or lipidic signals
    • Line 230 and others: IT IS instead of IT’S (which is informal English) Also apply to don’t à do not, etc (throughout the text, line 269, etc)
    • Line 231, 237, figure 3 legend, and many other places: reactions do not happen “thanks” to enzymes. Correct
    • Line 248: could happen à occurs…
    • Line 260: non-cannabinoid
    • Figure 4 legend: Anandamide (AEA)
    • Line 274: PPAR alpha was not mentioned before, spell it out
    • Line 290: deduced instead of deducted
    • Various lines (like 300 but many others): missing space between a stop and next word
    • General: in science data or information is not “noticed”, it is reported, described, etc
    • Line 345: “the pathology” , which one?
    • A high-fat diet instead of AN high
    • Diet-induced instead of diet induced
    • Line 430: remove “of course”

Author Response

We thank reviewer 1 for all the recommendations. We found them helpful for improving the manuscript.

  • QUERY 1: Introduction: Obesity is introduced without mentioning at first why it is considered a chronic pathology. There is also an underevaluation of other factors very important in the development of obesity and overvaluation of food addiction; for instance, there is great relevance of what we eat rather than the density of calories, including process foods with certain types of fats and sugars, or also there is no mention regarding the importance of inflammation at all. As well there are many cases of what is known as “healthy obese” which are not mentioned in the introduction.

ANSWER: Following your advice, in the introduction we added a part about obesity being a chronic pathology (line 42) and the importance of inflammation in obesity (line 81-92). Also, we mentioned the case of healthy obesity as you suggested, in lines 93-102. Factors influencing the development of obesity were listed in line 52-57, and mention to processed foods was done in line 62.

  • QUERY 2 Authors have used primarily references of reviews instead of original work. References must be changed to include the original work that has been done, which sometimes is also just 1 or 2 references no more. When not feasible for references limitations, then state “reviewed in XX”, and try using the review from the people that described it

ANSWER: As Referee recommended, further bibliography has been added to all the review, citing both, the original work and the reviews published when possible

  • QUERY 3 Authors are sometimes biased towards their field: they consider that very few studies describe the role of the ECS in feeding while they state that CB1 is in the mitochondria, although there are very few studies that have shown it.

ANSWER: We understand referee’s concern on the potential overstatement of mitochondrial CB1 cannabinoid receptors for explaining the metabolic changes associated to dysregulation/pharmacological manipulation of the endocannabinoid system. Our original intention was to indicate that there is a limited availability about studies linking endocannabinoid system expression in the brain and obesity.  In fact, we center in those studies on brain mitochondrial expression of CB1 receptors because of its mechanistic relevance for bioenergetic control.  (Our group has also described its presence in both, striatal muscle and cardiomyocites’ mitochondira). However, following your advice, we added in line 237 that only some studies described CB1 expression in mitochondria, underlying the need to a more in depth analysis of brain CB1 receptors in relation with obesity, especially in brain circuits relate to motivational and homeostatic processes dealing with food intake/energy expenditure.

  • QUERY 4: Table 1: It is not possible to distinguish between different rows. It misses novel compounds with double activity over CB1 and CB2, for example (+)-trans-Cannabidiol-2-hydroxy pentyl.. The title of table 1 does not reflect its content: there is no mechanism of action in the table. It would, in fact, be more informative if another column is added to include the impact regarding food intake/addiction/obesity of the different targeting strategies mentioned in the first column.

ANSWER: In table 1, different rows have been evidenced. (+)-trans-Cannabidiol-2-hydroxy pentyl was added in description, as  suggested  (line 676). Also the title of the table was changed, following reviewer’s recommendations.

  • QUERY 5: Authors talk about palatability but do not mention that the receptors are expressed in the taste buds, regulating taste.

ANSWER: A sentence about taste buds and CB1 receptor expression was added in line 467, and a small paragraph about RIO studies was added in lines 121-131, 421-422.

  • QUERY 6 Line 456: authors do not really mention periphery before. Line 459: it was not known that CB1 antagonism could have an impact on metabolism through its action on peripheral tissues before rimonabant. Indeed, it was after the RIO study that benefits independently of food intake were noticed.

ANSWER: Now, all compounds cited in the review are accompanied by a reference on their original study, where it’s possible to find compound specificity towards CB1 receptor. For each compound, in the work we only cited its most important activity.

  • QUERY 7: To the knowledge of this reviewer, JD5037 is not moving into a further clinical trial. If authors have more information regarding that, please include a reference

ANSWER: Reference about JD5037 moving to further clinical trial was found in “Resat Cinar, Malliga R. Iyer, George Kunos, The therapeutic potential of second and third generation CB1R antagonists, Pharmacology & Therapeutics, Volume 208, 2020, 107477, ISSN 0163-7258, https://doi.org/10.1016/j.pharmthera.2020.107477”. No further reference was found about this, so the information was removed from the work.

  • QUERY 8: Why do the authors talk about peripherally restricted compounds if the point of the review was to focus on the CNS and addiction? And what are the implications of the double-acting compounds (such as double CB1-iNOS) regarding food addiction?

ANSWER: A phrase justifying our choice to talk also about peripherally restricted compounds was added in lines 592-598; while the implications of using double acting compounds were clarified in lines 658-659.

  • QUERY 9: Line 601: The ECS has been studied for the treatment of complications associated with obesity.
  • Line 602/603: not all of them neither the majority of them regulate appetite, nor the reward response. So they are not all perfect candidates to treat food addiction. There is overall a missing message: which compounds (to my understanding CNS penetrant?) would be good candidates for the treatment of food addiction, and which are the cons of using a brain-penetrant blocker of CB1? Do the authors expect a different outcome than Rimonabant? And why? This is very important from both the scientific and the pharmacological and clinical points of view. Authors must include the risks of this strategy and a risk management plan. Also, the results obtained (metabolically) by using synthetic cannabinoids that bypass the CNS in obesity are on many occasions very good. In the current review, the importance, if any, of targeting the CNS is missing in the message, and needs to be highlighted
  •  

ANSWER: Unfortunately, we can’t state what drugs may be candidates to treat food addiction, since few studies have investigated the role of the cited molecules in reward, as stated in lines 715-717. Additionally, in lines 722-737 we explain why it’s important to target CNS. No evaluation of the risks of the strategy or a risk management plan has been added, since it goes beyond the aim of this review, but in line 733-735 we underline the importance of clinical trials.

QUERY 10: The manuscript needs to be deeply proofread for typos and also checked by an English native/high-level speaker as it is full of mistakes, sometimes misleading the message. 

ANSWER: As reviewer suggested, the manuscript underwent MDPI English editing, and all typos noticed were removed.

Reviewer 2 Report

The review titled “Obesity as a condition determined by food addiction: should brain endocannabinoid system alterations be the cause and its modulation the solution?” describes recent relevant studies on endocannabinoid system and its role in the central control of food intake and food addiction, with particular attention to the endocannabinoid system as a pharmacological target for obesity and food addiction. Authors performed a very good summary, and the paper is well written. However, there are important points that should be mentioned and a couple of references that need to be added.

Specific recommendations are detailed below.

At page 9 line 336 Authors report that the obesity is characterized by an overactive endocannabinoid system with increased plasma levels of AEA and 2-AG. Please add more references at the end of the statement, for example:

- Silvestri, C. & Di Marzo, V. The endocannabinoid system in energy homeostasis and the etiopathology of metabolic disorders. Cell Metab 2013; 17, 475–490.

-  Quarta C, Mazza R, Obici S, Pasquali R, Pagotto U. Energy balance regulation by endocannabinoids at central and peripheral levels. Trends Mol Med 2011; 17: 518–526.

-  Bermudez-Silva FJ, Cardinal P, Cota D. The role of the endocannabinoid system in the neuroendocrine regulation of energy balance. J Psychopharmacol 2012; 26: 114–124.

-  DiPatrizio NV, Piomelli D. The thrifty lipids: endocannabinoids and the neural control of energy conservation. Trends Neurosci 2012; 35: 403–411.

- Blüher M, Engeli S, Klöting N, et al. Dysregulation of the peripheral and adipose tissue endocannabinoid system in human abdominal obesity. Diabetes. 2006; 55:3053–60

- Engeli S, Böhnke J, Feldpausch M, et al. Activation of the peripheral endocannabinoid system in human obesity. Diabetes. 2005; 54:2838–43

- Sipe JC, Scott TM, Murray S, et al. Biomarkers of endocannabinoid system activation in severe obesity. PLoS One. 2010;20(1): e8792

One important point that should be mentioned is the fact that the changes in the plasma endocannabinoid level may also be influenced by the quality of the dietary fat. Furthermore, it is known that obese subjects have a chronically disrupted lipid metabolism and/or an unbalanced dietary regimen which in turn may influence endocannabinoid biosynthesis. Please see reference:

- Banni, S. & Di Marzo, V. Effect of dietary fat on endocannabinoids and related mediators: consequences on energy homeostasis, inflammation and mood. Mol Nutr Food Res 2010; 54, 82–92.

Besides, an unbalanced dietary regimen correlated with the endocannabinoid system could be due to variations in hunger/satiety and energy balance which is modulated genetically and by taste sensitivity. A paragraph describing this field and relative references must be included.  Please see the following references:

  • Tomassini Barbarossa, I.; et al. Taste sensitivity to 6-n-propylthiouracil is associated with endocannabinoid plasma levels in normal-weight individuals. Nutrition 2013, 29, 531–536
  • Carta, G., et al. (2017). Participants with normal weight or with obesity show different relationships of 6-n-propylthiouracil (PROP) taster status with BMI and plasma endocannabinoids. Rep. 7:1361.
  • Melis M, et al. (2017) Polymorphism rs1761667 in the CD36 Gene Is Associated to Changes in Fatty Acid Metabolism and Circulating Endocannabinoid Levels Distinctively in Normal Weight and Obese Subjects. Front. Physiol. 8:1006

Please pay attention to punctuation, in different parts of the text Authors use inappropriately the semicolon ( ; ): for example at page 1 line 42  “other factors; such as gender”.  Please correct it to a colon ( : ).

At page 3 line 107 Authors state: “genetic conditions; differently from drug” or at line 129 Authors state “binge/intoxication; withdrawal/negative affect and…” please replace the semicolon with a comma in these two cases. Please check the punctuation throughout the text.

Author Response

We thank the reviewer for the suggestions.

QUERY 1: At page 9 line 336 Authors report that the obesity is characterized by an overactive endocannabinoid system with increased plasma levels of AEA and 2-AG. Please add more references at the end of the statement, for example:

Part of the suggested references have been added to the manuscript, in particular the ones form Silvestri et al, Di Patrizio et al, Engeli et al, Sipe et al; that can be found in line 688.

QUERY 2: One important point that should be mentioned is the fact that the changes in the plasma endocannabinoid level may also be influenced by the quality of the dietary fat. Furthermore, it is known that obese subjects have a chronically disrupted lipid metabolism and/or an unbalanced dietary regimen which in turn may influence endocannabinoid biosynthesis:

Also, we agree with the suggestion to mention that diet can influence endocannabinoid levels. A small paragraph about this has been added in lines 695-699.

QUERY 3: Besides, an unbalanced dietary regimen correlated with the endocannabinoid system could be due to variations in hunger/satiety and energy balance which is modulated genetically and by taste sensitivity. A paragraph describing this field and relative references must be included

We added also some lines describing how endogenous levels of endocannabinoids may be influenced by altered lipid metabolism and taste sensibility, in lines 699-705.

QUERY 4: Please pay attention to punctuation, in different parts of the text Authors use inappropriately the semicolon ( ; )

Punctuation has been revised throughout the text.

Reviewer 3 Report

The authors are questioning about Obesity as a condition determined by food addiction: should brain endocannabinoid system alterations be the cause and its modulation the solution?

Please see below my suggestion in order to improve both the shape and content of the manuscript.

English must be moderate revised.

Please remove empty L37.

In the entire manuscript please add an empty space before each reference inserted in text (the square brackets containing the reference should not be attached to the word before it). Please revise and correct the entire manuscript in this regard.

Please add and develop the AIM of your research as the last, separate paragraph of the Introduction section. In the actual form of the manuscript it is totally missing. Why you chose this topic? What novelty character you research has? What special aspects it presents? etc.

2. Food addiction: Please remove : Never adding : after a title; same for all sections 3 to 6.

Figure 1. must be inserted in the text after the text paragraph mentioning it, namely after reference [29] (not in section 3). Moreover, it is blurred - please replace it with a better quality figure. According to the Instructions for authors, please explain all abbreviations used in the figure, under it, after its title and explanation (L199).

L151. ...in figure 1 we can see...please replace with it can be seen /observed. Use the impersonal manner of addressing instead of personal one. It sounds more professional. Revise the entire manuscript in this regard.

Please remove text L176-177. As I mentioned before, aim of the study must be inserted as the last paragraph of 1st section.

Figure 2 has unreadable parts. Explain also the abbreviations under it. Please adjust it size to be easier readable (just check the size of the text used for the title in Figure 1 and it compare with size of the text inserted in the rectangles in Figure 2) - you will understand my point of view. Make them appropriate and easy readable.

Figures 3 and 4. Blurred and unreadable. Please replace them. Explain also the abbreviations under each of them. 

Please unbold the text in the manuscript: L462, 485, 489, 496, 499-501, etc. Revise the entire manuscript in this regard.

All genes must be written in Italics.

In Discussion part. Please further detail the role of leptin in obesity and if new advances have been done regarding the leptin resistance at the cerebral level. Please check: Exp Ther Med. 2020 Jul;20(1):121-128. doi: 10.3892/etm.2020.8663. Epub 2020 Apr 15. PMID: 32509004; PMCID: PMC7271710. Please detail the efficiency of the analyzed drugs in the reduction of obesity, and discuss the role of GLP-1 in modulating the endocannabinoid system and the role of GLP-1 agonists in this context. Furthermore, please discuss if endocannabinoid system is implied directly in other pathologies such as cardiovascular disease and if drugs that modulate the endocannabinoid system increase the risk for this comorbidity Please check and refer to Stoicescu M. et al. The role of increased plasmatic renin level in the pathogenesis of arterial hypertension in young adults. Rom J Morphol Embryol. 2011;52(1 Suppl):419-23. and Moisi, M.I.; et al. Framing Cause-Effect Relationship of Acute Coronary Syndrome in Patients with Chronic Kidney Disease. Diagnostics 2021, 11, 1518. https://doi.org/10.3390/diagnostics11081518

Table 1. All abbreviations used in the table must be detailed/explained in full under the table, according to the Instructions for authors which state:

  • Acronyms/Abbreviations/Initialisms should be defined the first time they appear in each of three sections: the abstract; the main text; the first figure or table. When defined for the first time, the acronym/abbreviation/initialism should be added in parentheses after the written-out form.

L621. Please remove "In conclusion, we can state that". It is repetitive In conclusion (the text is in Conclusions section) and "we can state" because you must avoid personal manner of addressing and use the impersonal one which sounds more professional.

L621. not the number of obesity is growing every day, but the number of subjects (please add of subjects).

References must be written in the MDPI style, according to the Instructions for authors. Please check and apply https://www.mdpi.com/journal/pharmaceuticals/instructions . Please provide ALL the data required for each reference, including the data of first 10 authors et al. (this is the number of authors for whom information are needed)

Author Response

We thank the reviewer for all the suggestions.

QUERY 1: English must be moderate revised.

Following your advice, the manuscript underwent MDPI English editing service.

QUERY 2: Please remove empty L37; In the entire manuscript please add an empty space before each reference inserted in text (the square brackets containing the reference should not be attached to the word before it). Please revise and correct the entire manuscript in this regard; Food addiction: Please remove : Never adding : after a title; same for all sections 3 to 6

Line 37 was removed and all the references have been separated with an empty space from the word before. All : in sections 2-6 were removed.

QUERY 3: Please add and develop the AIM of your research as the last, separate paragraph of the Introduction section. In the actual form of the manuscript it is totally missing. Why you chose this topic? What novelty character you research has? What special aspects it presents? etc.

As you suggested, a small paragraph containing the aim of the study was added in lines 170-176.

QUERY 4: Figure 1. must be inserted in the text after the text paragraph mentioning it, namely after reference [29] (not in section 3). Moreover, it is blurred - please replace it with a better quality figure. According to the Instructions for authors, please explain all abbreviations used in the figure, under it, after its title and explanation (L199); Figure 2 has unreadable parts. Explain also the abbreviations under it. Please adjust it size to be easier readable (just check the size of the text used for the title in Figure 1 and it compare with size of the text inserted in the rectangles in Figure 2) - you will understand my point of view. Make them appropriate and easy readable. Figures 3 and 4. Blurred and unreadable. Please replace them. Explain also the abbreviations under each of them

We agreed with the suggestion of moving figure 1, moreover all figures were changed and replaced with better quality figures and the abbreviations in each figure were explicated in figure description.

QUERY 5: L151. ...in figure 1 we can see...please replace with it can be seen /observed. Use the impersonal manner of addressing instead of personal one. It sounds more professional. Revise the entire manuscript in this regard. Please remove text L176-177. As I mentioned before, aim of the study must be inserted as the last paragraph of 1st section:

All the manuscript was changed to the impersonal manner as you suggested for line 151 and text in line 176-177 was cancelled.

QUERY 6: Please unbold the text in the manuscript: L462, 485, 489, 496, 499-501, etc. Revise the entire manuscript in this regard

Bold in lines 550-720 was used to evidence the name of the molecules discovered, and was used only the first time the molecule was cited.

QUERY 7: All genes must be written in Italics.

No gene name was written in the manuscript.

QUERY 8: In Discussion part. Please further detail the role of leptin in obesity and if new advances have been done regarding the leptin resistance at the cerebral level. Please check: Exp Ther Med. 2020 Jul;20(1):121-128. doi: 10.3892/etm.2020.8663. Epub 2020 Apr 15. PMID: 32509004; PMCID: PMC7271710. Please detail the efficiency of the analyzed drugs in the reduction of obesity, and discuss the role of GLP-1 in modulating the endocannabinoid system and the role of GLP-1 agonists in this context. Furthermore, please discuss if endocannabinoid system is implied directly in other pathologies such as cardiovascular disease and if drugs that modulate the endocannabinoid system increase the risk for this comorbidity Please check and refer to Stoicescu M. et al. The role of increased plasmatic renin level in the pathogenesis of arterial hypertension in young adults. Rom J Morphol Embryol. 2011;52(1 Suppl):419-23. and Moisi, M.I.; et al. Framing Cause-Effect Relationship of Acute Coronary Syndrome in Patients with Chronic Kidney Disease. Diagnostics 2021, 11, 1518. https://doi.org/10.3390/diagnostics11081518

In discussion part, we followed your advice about adding evidence of relationship between GLP-1 and endocannabinoid system, as you can see in lines 571-577. However, we did not discuss the role of the endocannabinoid system in cardiovascular diseases, since it goes beyond the aim of this review. For the same reason we did not discuss leptin resistance in obesity.

QUERY 9: Table 1. All abbreviations used in the table must be detailed/explained in full under the table, according to the Instructions for authors; Please remove "In conclusion, we can state that".

As you suggested, we explained all the abbreviations of table 1, removed the “in conclusion” appearing in line 621 and changed the phrase in the same line.

QUERY 10: References must be written in the MDPI style, according to the Instructions for authors. Please check and apply https://www.mdpi.com/journal/pharmaceuticals/instructions

References were adjusted to MDPI style, according to your recommendation.

Round 2

Reviewer 1 Report

I would like to congratulate the authors; wonderful work and a beautiful review.

Minor change: in line 600 "It is of special relevance that all the CB1 cannabinoid receptors antagonists" CB1 has the word "cannabinoid" in it, as defined in line 283.

Reviewer 3 Report

All the requirements have been fulfilled. I reccomend publication.